# Systematic characterization of plant-associated bacteria that can degrade indole-3-acetic acid

Lanxiang Wang[1], Yue Liu[1,2], Haoran Ni[1], Wenlong Zuo[1], Haimei Shi[1], Weixin Liao[1], Hongbin Liu[1], Jiajia Chen[3], Yang Bai[4], Hong Yue[5], Ancheng Huang[6], Jonathan Friedman[7], Tong Si[1], Yinggao Liu[2], Moxian Chen[3]*, Lei Dai[1]*

1 CAS Key Laboratory of Quantitative Engineering Biology, Shenzhen Institute of Synthetic Biology, Shenzhen Institutes of Advanced Technology, Chinese Academy of Sciences, Shenzhen, China, 2 National Key Laboratory of Wheat Improvement, College of Life Science, Shandong Agricultural University, Taian, China, 3 State Key Laboratory of Green Pesticide, Key Laboratory of Green Pesticide and Agricultural Bioengineering, Ministry of Education, Center for R&D of Fine Chemicals of Guizhou University, Guiyang, China, 4 State Key Laboratory of Plant Genomics, Institute of Genetics and Developmental Biology, The Innovative Academy of Seed Design, Chinese Academy of Sciences, Beijing, China, 5 State Key Laboratory of Crop Stress Biology in Arid Areas, College of Agronomy, Northwest A&F University, Xianyang, China, 6 Shenzhen Key Laboratory of Plant Genetic Engineering and Molecular Design, SUSTech-PKU Institute of Plant and Food Science, Department of Biology, School of Life Sciences, Southern University of Science and Technology, Shenzhen, China, 7 Institute of Environmental Sciences, Hebrew University, Rehovot, Israel

☯ These authors contributed equally to this work.
* cmx2009920734@gmail.com (MXC); lei.dai@siat.ac.cn (LD)

**Data Availability Statement:** The RNA-Seq data in this study were deposited to NCBI with BioProject number is PRJNA1164981. Whole genome sequences of 183 isolates investigated in this

## Abstract

Plant-associated microbiota affect pant growth and development by regulating plant hormones homeostasis. Indole-3-acetic acid (IAA), a well-known plant hormone, can be produced by various plant-associated bacteria. However, the prevalence of bacteria with the capacity to degrade IAA in the rhizosphere has not been systematically studied. In this study, we analyzed the IAA degradation capabilities of bacterial isolates from the roots of *Arabidopsis* and rice. Using genomics analysis and in vitro assays, we found that 21 out of 183 taxonomically diverse bacterial isolates possess the ability to degrade IAA. Through comparative genomics and transcriptomic assays, we identified iac-like or iad-like operon in the genomes of these IAA degraders. Additionally, the putative regulator of the operon was found to be highly conserved among these strains through protein structure similarity analysis. Some of the IAA degraders could utilize IAA as their carbon and energy source. In planta, most of the IAA degrading strains mitigated *Arabidopsis* and rice seedling root growth inhibition (RGI) triggered by exogenous IAA. Moreover, RGI caused by complex synthetic bacterial community can be alleviated by introducing IAA degraders. Importantly, we observed increased colonization preference of IAA degraders from soil to root according to the frequency of the biomarker genes in metagenome-assembled genomes (MAGs) collected from different habitats, suggesting that there is a close association between IAA degraders and IAA producers. In summary, our findings further the understanding of the functional diversity and potential biological roles of plant-associated bacteria in host plant root morphogenesis.

study, the code and processed data for graphs in the figures are bundled in the Github [https://github.com/NHR326/Indole-3-acetic-acid-degradation], Zenodo [DOI: 10.5281/zenodo.13990203], and S6 Table.

**Funding:** National Natural Science Foundation of China (Grant number. 32061143023). Received by LD. Guangdong Basic and Applied Basic Research Foundation (Grant number. 2023A1515012006). Received by LXW. Central Government Guides Local Science and Technology Development Fund Projects (Grant number. Qiankehezhongyindi (2024) 007). Received by MXC. Guizhou Provincial Basic Research Program (Grant number. (Natural Science)-ZK[2023]-099). Received by MXC. Program of Introducing Talent to Chinese Universities (Grant number. 111 Program-D20023). Received by MXC. The funders had no role in study design, data collection and analysis, decision to publish, or preparation of the manuscript.

**Competing interests:** The authors have declared that no competing interests exist.

**Abbreviations:** ABA, abscisic acid; CFU, colony-forming unit; DEG, differentially expressed gene; IAA, indole-3-acetic acid; LC-MS, liquid chromatography-mass spectrometry; MAG, metagenome-assembled genome; MarR, multiple antibiotic resistance regulator; MRM, multiple reaction monitoring; PGPR, plant growth-promoting rhizobacteria; RGI, root growth inhibition; ROS, reactive oxygen species; TM, template modeling; TSB, tryptic soy broth; WGS, whole genome sequence.

## Introduction

Indole-3-acetic acid (IAA) is a typical auxin naturally produced by plants, playing a crucial role in various aspects of plant growth and development, such as cell division, elongation, and differentiation [1–4]. IAA is primarily synthesized in developing plant tissues and highly concentrated in the root's apical part, where the organizing quiescent center accumulates a distinct IAA concentration gradient [5,6]. Additionally, cells in the root apex exhibit a highly active capacity for IAA synthesis [6]. Alongside root cell exfoliation and other transportation strategies, considerable levels of IAA were detected in root exudates [7–9].

The surface and internal parts of plants are colonized by millions of commensal microorganisms, collectively known as the plant-associated microbiome [10]. Among these habitats, the root is one of the most critical, with numerous studies suggesting that root exudates, such as plant hormones, significantly influence the structure and function of the root-associated microbiome (predominantly the rhizosphere microbiome) [11–13]. Over millions of years of coexistence with their hosts, microbes have evolved multiple strategies for colonization, including the ability to synthesize and catabolize plant-specific metabolites, such as IAA [14]. It is estimated that 80% of commensal microorganisms isolated from the rhizosphere can produce IAA [15]. However, the proportion of the microbes possessing IAA degradation pathways in the rhizosphere is unknown, not to mention the potential ecological roles of these microbes in their habitats.

Two main pathways for IAA consumption among aerobic bacteria have been characterized (S1 Fig). The gene cluster *iacABCDEFGHIR* (hereafter iac-like operon), responsible for IAA catabolism into catechol, was identified in *Pseudomonas putida* 1290, *Paraburkholderia phytofirmans* PsJN, *Acinetobacter baumannii*, *Enterobacter soli*, and *Caballeronia glathei* [16–22] (S2 Fig). In contrast, the IAA degradation locus *iadABCDEFGHIJKLMNR* (hereafter iad-like operon), resulting in anthranilate as the end-product, was identified in *Variovorax paradoxus* CL14, *Achromobacter*, and *Bradyrhizobium japonicum* [23–28] (S2 Fig). Moreover, among these components, heterologous expression and gene knock-out experimental validations suggested that *iacAE* or *iadDE* are necessary for IAA bio-transformation in *C. glathei* and *V. paradoxus*, respectively [22,24]. Besides genes encoding enzymes responsible for IAA catabolism, there are components responsible for regulating operon expression in the cluster. The current studies on expression regulation of the operon suggest that most of the iac-like and iad-like operons contain a MarR (multiple antibiotic resistance regulator) family regulator [21,24,29]. For example, the crystal structure and binding properties of IadR, a member of the MarR family, were recently determined in *V. paradoxus*, revealing the mechanism of operon expression regulation [24].

Here, through combining comparative genomics, transcriptomics, with in vitro degradation assay, we systematically evaluated the IAA degradation capacity among 183 bacterial strains which were isolated from *Arabidopsis* (*Arabidopsis thaliana*) and rice (*Oryza sativa*) roots. We predicted the IAA-degrading candidates based on the presence of iac-like operon or iad-like operon in their genomes, followed with experimental validation. We identified 21 strains belonging to 7 genera, including 2 previously unreported genera, *Sphingopyxis* and *Curvibacter*, that typically exhibit bona fide IAA degradation activity. All IAA degrading strains carry the iac-like or iad-like operon, and the transcriptomic results show that iac and iad gene clusters were up-regulated by IAA stimulation. Moreover, the MarR family regulator was present in the operon of all genera with high similarity in their protein structures except for *Pseudomonas*, which had a putative two-component regulatory system in their iac-like operons. Furthermore, in subsequent assays, we found that some of the IAA degraders could utilize IAA as their sole carbon and energy source. In planta, our results demonstrated that

exogenous IAA and complex synthetic bacterial community induced primary root growth inhibition (RGI) was disrupted by some of the IAA degraders, suggesting an important role of IAA degraders in the rhizosphere for host plant growth and development. Finally, by analyzing metagenome-assembled genomes (MAG) and whole genome sequences (WGS) of microbial isolates from different habitats, we found that the prevalence of IAA degraders is positively correlated with naturally occurring IAA resources.

## Results

### Genomic analysis and experimental validation were employed to screen for IAA degraders

Previous reports have identified 2 types of aerobic auxin catabolic gene clusters in bacteria, known as iac-like and iad-like operons [24]. Key genes *iacAE* and *iadDE* play essential roles in IAA degradation [22,24]. To systematically evaluate IAA degradation capabilities in bacterial commensals isolated from *Arabidopsis* and rice root, we profiled loci containing genes homologous to iac-like and iad-like operons by scanning 183 bacterial genomes from our laboratory collection (Figs 1 and S3 and S1 Table). A total of 21 strains containing iac-like operon, or iad-like operon, were identified as IAA-degrading candidates (S3 Fig). Among these, predicted proteins encoded by the genomes of 4 *Pseudomonas* strains, 5 *Acinetobacter* strains, and 1 *Curvibacter* strain show high amino acid sequence identity to the reported IacA and IacE (identity >60%). Additionally, predicted proteins encoded by the genomes of 4 strains belonging to *Variovorax* and 3 to *Achromobacter*, exhibit high amino acid sequence identity to the experimentally validated IadD and IadE. A strain belonging to *Sphingopyxis* harbors genes encoding predicted proteins display moderate amino acid sequence identity to the IadD and IadE (identity between 40% and 60%). Interestingly, the genomes of 3 *Sphingomonas* strains contain both *iacA/E* and *iadD/E*, which encode the proteins with moderate amino acid sequence identity to the templates (Fig 1 and S2 Table).

To validate the predicted results, we tested the IAA degradation capability of 131 *Arabidopsis* root bacterial isolates and 52 rice root bacterial isolates using the colorimetric assay [30]. After 72 hours incubation, bacterial growth and the percentage of IAA degradation of the strains were measured and calculated. A total of 21 out of 183 strains displayed considerable IAA degradation efficiency, such that over 60% of the IAA content were consumed (Fig 1). To further detect the consumption of IAA by the IAA-degrading candidates, the supernatant of the culture was analyzed by a highly accurate analytical approach of liquid chromatography-mass spectrometry (LC-MS). The specific peak of the IAA was undetectable in these IAA-degrading candidates cultures, suggesting that IAA was degraded by these strains (S4 Fig). Ultimately, 21 IAA degraders belonging to 7 genera were confirmed through experimental validation, which are consistent with the bioinformatics prediction.

### IAA degraders possess the iac/iad-like operon

To clarify and characterize the IAA catabolism gene clusters in our screened IAA degraders, we performed a BLASTp search using the amino acid sequences of the full iac and iad operons obtained from previous reports against the genomes of 21 strains (S2 Fig and S2 Table). A complete iac-like or iad-like operon was identified from the genomes of all the IAA degraders except for Root154_*Sphingopyxis* (Fig 2). Consistent with previous reports, strains from *Achromobacter* and *Variovorax* possess the iad-like operon, and strains from *Pseudomonas*, *Acinetobacter*, and *Sphingomonas* contain the iac-like operon in their genomes. For SE9_*Curvibacter* and Root154_*Sphingopyxis*, 2 novel identified IAA-degrading genera, an iac-like operon and a

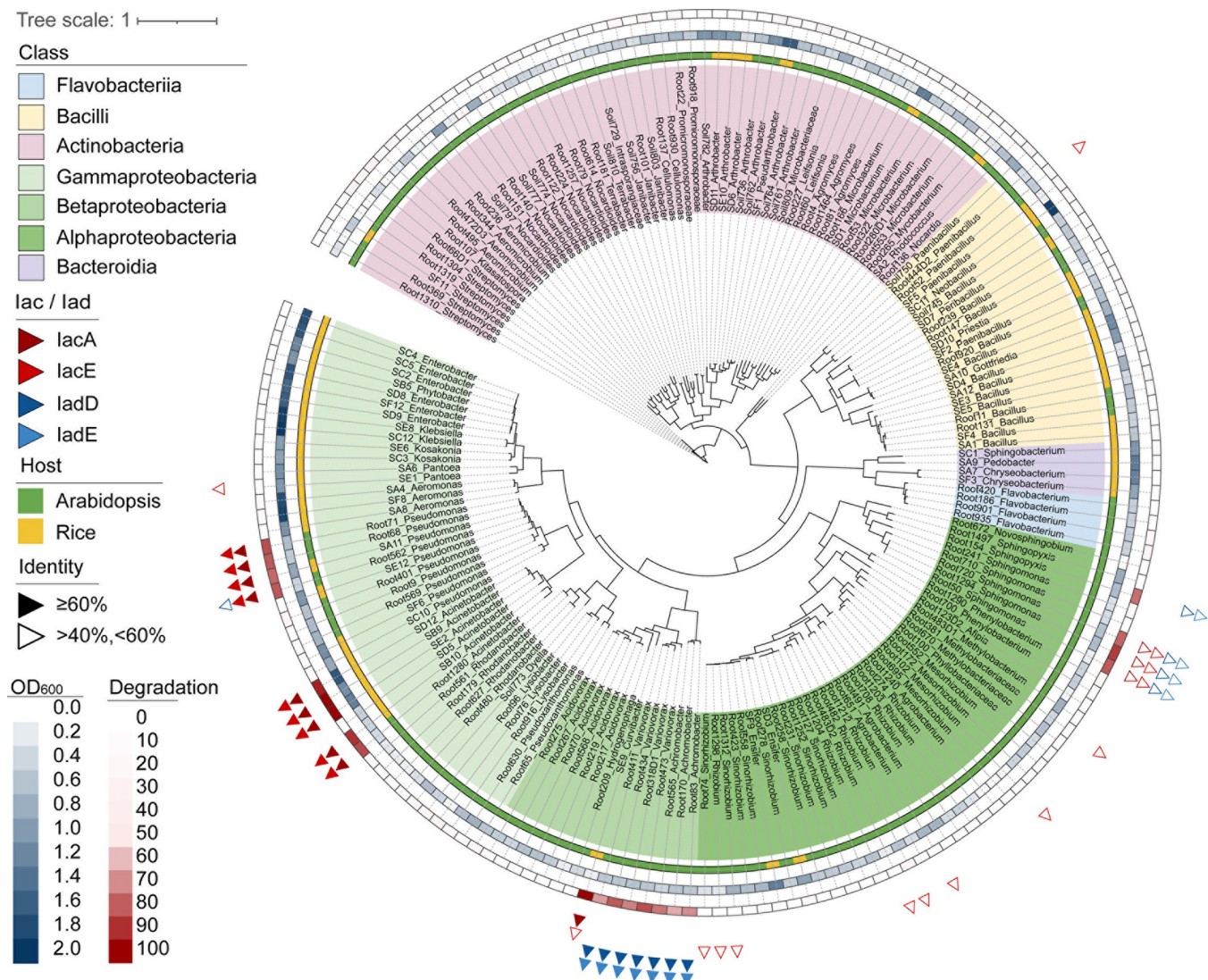

**Fig 1. IAA degradation genes were annotated in bacterial strains isolated from *Arabidopsis* and rice rhizosphere.** The WGS-based phylogenetic tree of the 183 strains was generated with phylophlan and visualized with iTOL. Strains annotated to have IacA/E or IadD/E (over 40% identity and 60% coverage with template amino acid sequences, see Methods) were labeled with triangles. The inner ring with blue color represent bacterial biomass as measured by $OD_{600}$ after 72 hours' growth in $^1/_2$ TSB. The outer ring with red color represent the percentage of IAA degradation of individual strains after 72 hours' incubation in $^1/_2$ TSB medium performed by colorimetric assay. IAA, indole-3-acetic acid; TSB, tryptic soy broth; WGS, whole genome sequence.

fragmented iad-like operon are present in their genomes, respectively (Fig 2). Additionally, aside from the complete iac-like operon, 2 sets of fragmentary iad-like gene clusters were also found in the genomes of strains from *Sphingomonas* (S5 Fig). Intriguingly, there is another potential fragmented iad-like operon (iad-2) present in the genome of Root154_*Sphingopyxis*, and this operon has the same gene cluster arrangement with strains from *Spingomonas* (S5 Fig).

To investigate the evolutionary relationships among different IAA degraders, we constructed a phylogenetic tree using concatenated iac/iad-like operon amino acid sequences. Intriguingly, strain organization in this tree differed substantially from the phylogenetic relationships based on WGS (S6 Fig). For example, *Curvibacter* and other members of Burkholderiales exhibit closely related genomic evolutionary relationships, yet their gene clusters'

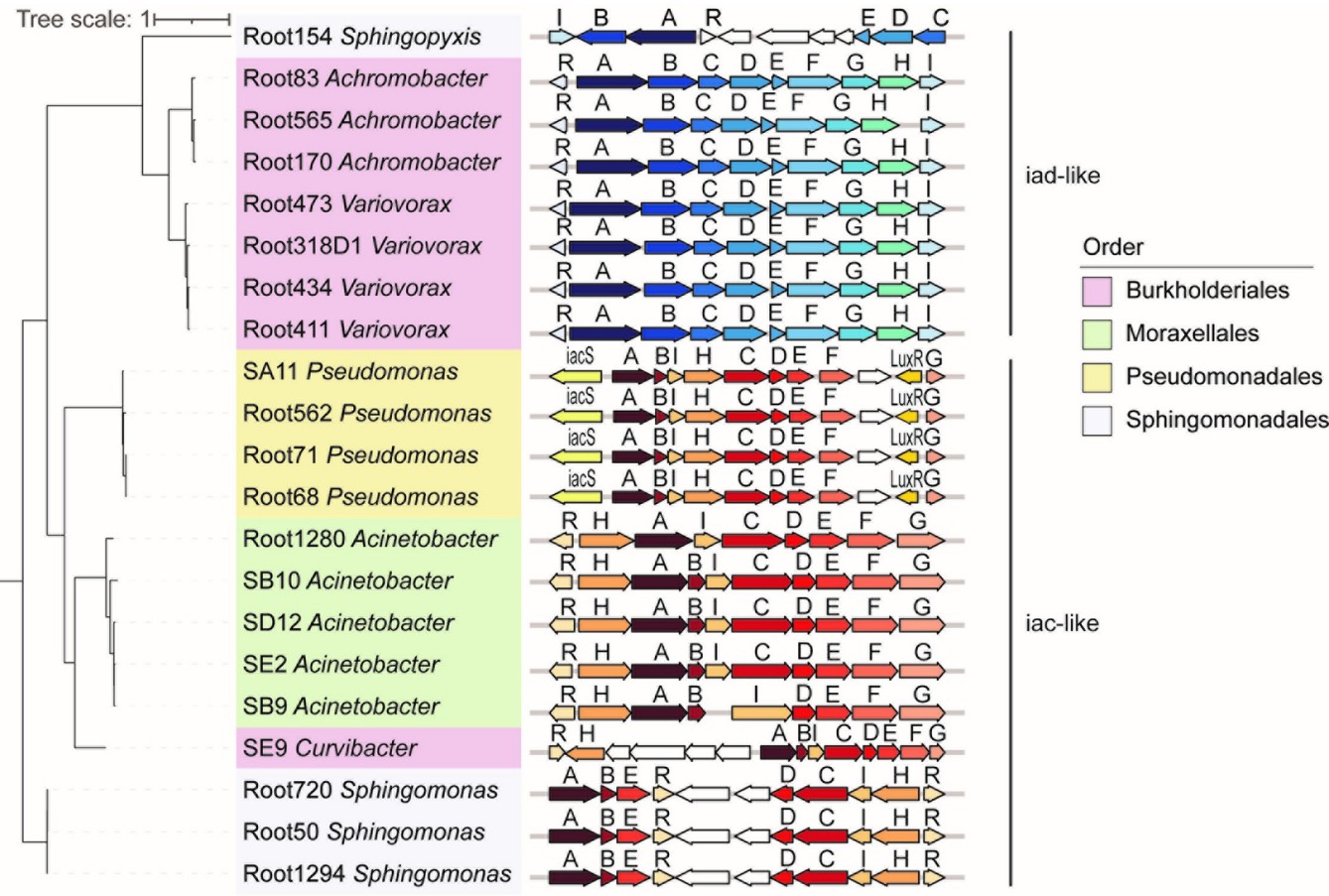

**Fig 2. Identification of gene clusters related to IAA degradation.** Organization of *iac* and *iad* gene clusters in 21 IAA-degrading strains. Phylogenetic tree constructed using the concatenated iac or iad gene cluster encoded amino acid sequences. Gene transcription directions are indicated by arrows. Letters above the red or blue arrows indicate function genes in the cluster with high protein sequence identity compared to the templates. White arrows indicate genes in this cluster with unknown function. IAA, indole-3-acetic acid.

evolutionary trajectories are divergent. Similarly, *Sphingomonas* and *Sphingopyxis*, both belonging to Sphingomonadales, display closely related genomic evolutionary relationships but occupy distinct branches on the phylogenetic tree based on functional gene clusters. Notably, *Sphingomonas* genomes contain iac-like operon, while the *Sphingopyxis* genome harbor an iad-like operon (Fig 2). These findings suggest that the evolutionary processes of these strains do not align perfectly with the acquisition of certain gene clusters. This discrepancy implies that the evolution of the microbes and the integration of these specific gene clusters into their genomes occurred at different rates or times, indicating a lack of synchronization between these 2 processes.

## The putative MarR family regulators exhibit high degree of structural conservation among IAA-degrading strains

In all IAA-degraders except *Pseudomonas*, both iac-like and iad-like operons contain 1 or 2 putative MarR-family regulator (R) that are involved in the operon expression regulation (Fig 2). In contrast to previous reports [24], our 4 *Pseudomonas* strains exhibited the involvement of a putative two-component regulatory system in their iac-like operons (Fig 2). Upon comparison with the iac operon in the related reference strain *Pseudomonas putida* 1290, it seems

that the components of the cluster may have been acquired from a distant genus, such as *Paraburkholderia phytofirmans* (S2 Fig).

Generally, the MarR regulator acts as a negative transcriptional factor for the operon since it binds to the upstream DNA region, inhibiting operon expression [18,19,24]. Due to the similar role of this protein in regulating iac and iad operon expression, we hypothesized that they are homologous with high amino acid sequence identity. To verify our hypothesis, multiple protein sequence alignment was performed using 21 potential MarRs identified from our 17 strains and 6 MarR reference proteins [24]. Consistent with previous reports, MarR regulators display considerable sequence diversity (S7 Fig). The phylogenetic tree based on the protein sequence of 27 MarRs demonstrated that IacR naturally separated from IadR which is consistent with previous reports [31]. Moreover, the 27 MarRs are divided into 3 distinct clusters (Fig 3A). MarRs are relatively conserved within a genus, while IacRs display more sequence diversity. It is worth noting that 3 strains from *Sphingomonas* contain 2 IacR encoding genes in their iac operons, respectively (Figs 2, 3, S7 and S8). Also, 2 MarRs were screened from Root154_*Spingopyxis* genome with low similarity, Root154_*Spingopyxis*_IadR1 (gene_00256), and Root154_*Spingopyxis*_IacR2 (gene_00334) (Figs 3A, S7 and S8). In addition, except for Root154_*Spingopyxis*_IadR1 (gene_00256), other MarRs from Sphingomonadaceae were grouped together and classified as IacR (Fig 3A).

To further elucidate the mechanism of functional conservation among MarRs, the predicted protein structures of 21 hypothetical MarRs were generated using AlphaFold2 [32] (Figs 3B and S8). The protein structures of 6 referential MarR proteins were retrieved from the PDB, and pairwise comparisons among the 27 MarRs were conducted by calculating their template modeling (TM) score [33,34]. A TM-score greater than 0.5 implies that 2 structures are likely to adopt the same fold and are evolutionary related [35]. As shown in the heatmap, all pairwise comparisons obtained a high TM-score (all pairwise TM-score > 0.58) (Fig 3 and S6 Table), suggesting that all MarRs displayed high similarity in their protein structures. In addition, the IAA binding sites of MarRs also exhibited high conservation, especially within a genus level (Fig 3A). The conservation of protein structure and the IAA binding sites may elucidate why different types of MarRs exhibit similar mechanisms of action in regulating the expression of downstream gene clusters.

## Iac- and iad-like operons were up-regulated by IAA

To explore how iac- or iad-like operons were regulated by IAA, we examined the transcriptomes of 5 strains in response to this compound. The strains selected for RNA-seq were Root71_*Pseudomonas*, Root170_*Achromobacter*, Root473_*Variovorax*, SE9_*Curvibacter*, and Root154_*Sphingopyxis*. They were cultured with M9 minimal medium supplemented with IAA or glucose, and cells were collected in specific time point (see Methods). Among them, Root154_*Spingopyxis* and SE9_*Curvibacter* were studied for IAA degradation activity for the first time. In total, 801, 492, 546, 1,264, and 701 differentially expressed genes (DEGs) were identified in these 5 strains, respectively (S3 Table). Both iac-like and iad-like operon were significantly up-regulated by IAA, except for Root154_*Sphingopyxis* (Fig 4). Furthermore, categorization of up- and down-regulated DEGs indicated that IAA treatment broadly influenced many metabolism pathways (S9 Fig).

It has been reported that chemicals in root exudates may enhance plant–microbe interactions through transcriptional regulation of bacterial motility [36]. Flagella are organelles used by microbes for movement. Notably, in Root473_*Variovorax* and Root154_*Sphingopyxis*, genes involved in flagellar biosynthesis, and chemotaxis proteins were up-regulated by IAA treatment when compared with glucose control, suggesting that for these 2 strains, IAA might

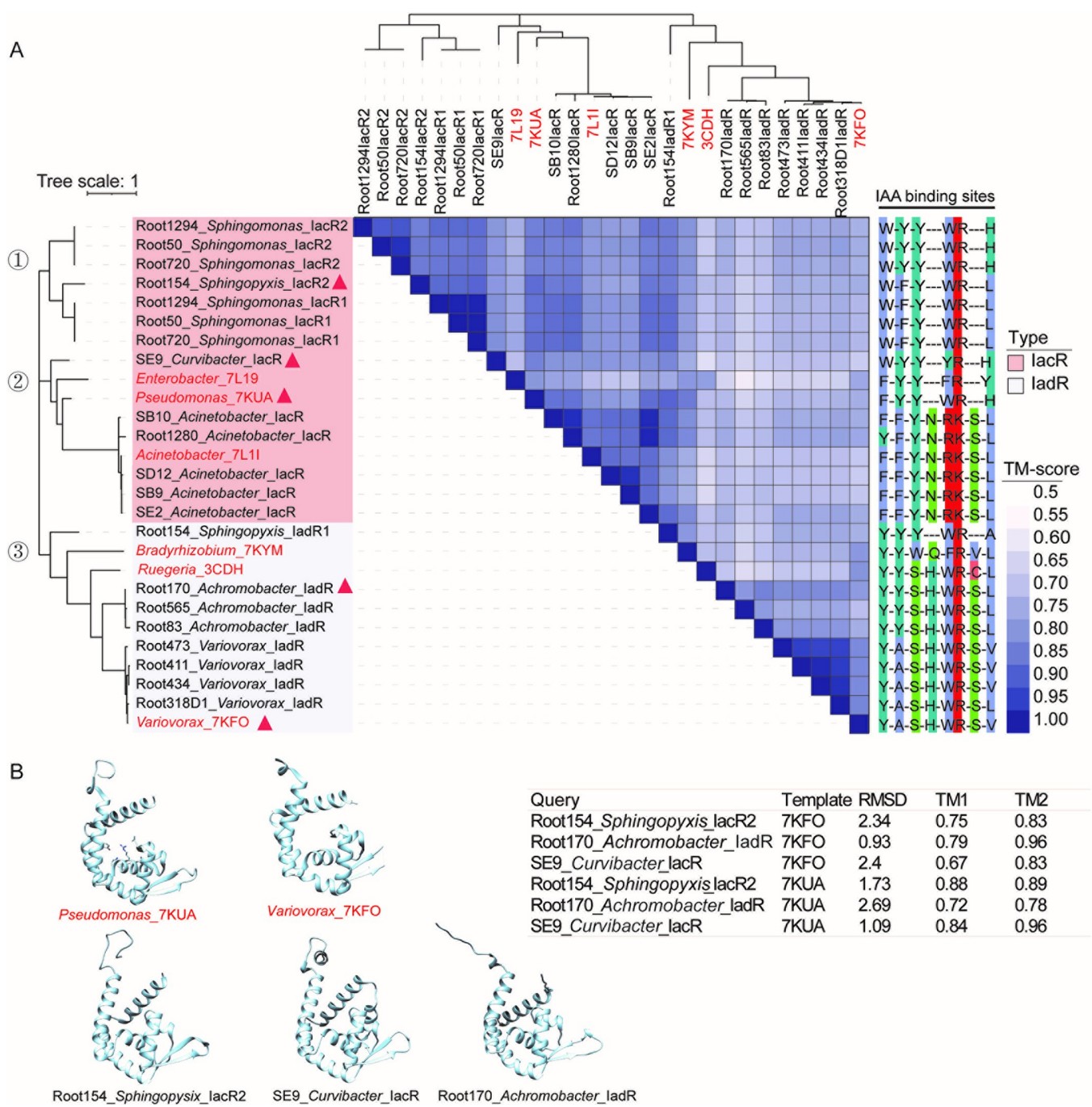

**Fig 3. Characterization of MarR protein structures.** (A) Protein structures of IacR or IadR (all belong to MarR family) are highly conserved among IAA-degrading strains. The heatmap displaying structural similarities among 27 MarR family proteins highlights the TM-score distribution (only the lower TM-score was displayed in the heatmap, all data was shown in S6 Table). IacR and IadR were retrieved from 17 IAA-degrading strains in this study, along with 6 reference proteins whose protein structures and IAA binding sites have been identified. The phylogenetic tree was constructed using the IacR or IadR amino acid sequences. The IAA binding sites of MarR were conservatively distributed in IacR or IadR. Six MarR templates used in this analysis (labeled in red) are *Acinetobacter baumannii*_7L1I, *Pseudomonas putida*_7KUA, *Enterobacter soli* ATCC BAA-2102_7L19, *Bradyrhizobium japonicum*_7KYM, *Ruegeria pomeroyi* DSS-3_3CDH, and *Variovorax paradoxus* CL14_7KFO. (B) Five selected representative structural models (marked with a red triangle on panel A), including 7UKA, 7KFO, and 3 MarR proteins from distinct clusters, are displayed. The TM-score and RMSD (root mean square deviation) between the query and template are shown in the table on the right. IAA, indole-3-acetic acid; MarR, multiple antibiotic resistance regulator; TM, template modeling.

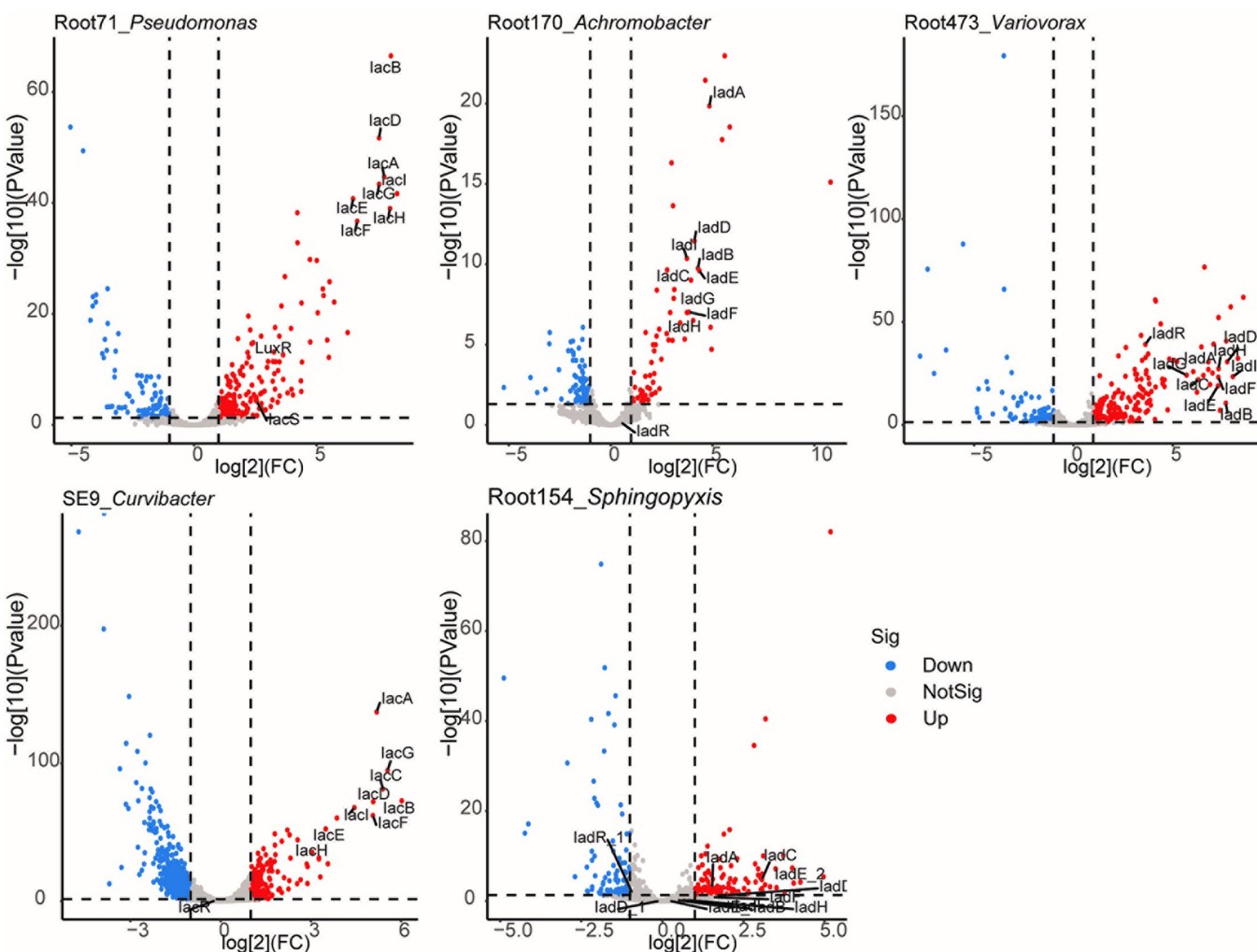

**Fig 4. RNA-seq reveals IAA-induced transcription of the iac-like or iad-like gene cluster.** The 5 selected strains for RNA-seq analysis were cultured with M9 medium supplemented with either IAA or glucose as the sole carbon source. The differential expression genes between IAA and glucose treatments were analyzed. Significantly up-regulated or down-regulated genes were identified with $\log_2^{\text{FoldChange}} \geq 1$ or $\leq -1$ and adjusted $p$-value $< 0.05$ as cutoffs. Three individual colony of each strain were used to perform the transcriptome analyses. Numerical values that underlie the data displayed in the volcano plots are in S6 Table. IAA, indole-3-acetic acid.

be more attractive than glucose. However, in Root71_*Pseudomonas* and Root170_*Achromobacter*, genes encoding flagellum-related DEGs and chemotaxis proteins were down-regulated by IAA stimulation, indicating that IAA elicited different molecular responses in different strains. Consistent with previous reports, catechol, the end-product of iac-pathway, will be further catalyzed by downstream enzymes, catABC [16,20,21]. Genes involved in catechol pathway in Root71_*Pseudomonas* and Root170_*Achromobacter* were up-regulated by IAA treatment (S3 Table, highlighted).

## IAA can be utilized as the carbon source

To evaluate the utilization of IAA by the IAA degraders, in vitro assays were performed among the screened 21 IAA degraders. The IAA degradation efficiency and bacterial growth of the strains were carried out in M9 minimal medium with exogenous IAA. In this study, 0.4 mM IAA with 0.05% ethanol for IAA solubilization was supplemented in the M9 minimal medium

as sole carbon source. Our results showed that strains from *Acinetobacter* and *Pseudomonas* presented the maximum degradation and growth rates, consuming IAA completely within 12 or 24 hours (Fig 5A and 5B). This effect may be attributed to the ability of *Acinetobacter* and *Pseudomonas* strains to utilize ethanol to support initial population expansion, followed by rapid IAA degradation (S10 Fig). Interestingly, the growth of *Acinetobacter* strains were slightly inhibited in the presence of IAA, whereas IAA accelerated the growth of *Pseudomonas* (S10 Fig). On the other hand, other strains exhibited slower growth in M9 medium, which consequently impacted their IAA degradation rates. Notably, strains such as Root473_*Variovorax*, Root50_*Sphingomonas*, and Root83_*Achromobacter* exhibited accelerated growth after metabolizing IAA within 24 hours, indicating their ability to utilize IAA as a sole carbon source for growth. Finally, strains from *Variovorax*, *Achromobacter*, *Curvibacter*, *Sphingomonas*, and *Sphingopyxis* exhibited either complete or partial IAA degradation within 72 hours, while also displaying extremely slow growth rate. This may suggest that the cell proliferation of these strains requires additional carbon and energy sources. The degradation of IAA observed in strains such as SE9_*Curvibacter* and strains from *Acinetobacter*, may serve alternative purposes, such as mitigating the toxicity of this compound, disrupting inter-microbial signaling, or facilitating their colonization on the root. Combined with the transcriptome analysis, our results here suggest that IAA triggers the expression of iac/iad-like operon and facilitates the biotransformation of this compound in the medium, even though IAA may not be suitable as sole carbon and energy source for cell growth.

## IAA degraders contribute to the regulation of plant root growth

Auxin homeostasis in plant roots is achieved through local synthesis, polar transport, and the contribution of IAA-producing/consuming microorganisms, which is crucial for root growth. To clarify the biological role of the root isolates possessing IAA degradation capability identified in this study, *Arabidopsis* seedlings and rice seedlings were transferred to half-strength MS agar medium supplemented with IAA and inoculated with the IAA degraders individually. After an extended growth period (7 days for *Arabidopsis* and 3 days for rice), primary root elongation was measured (see Methods). Inhibition of primary root growth (RGI) was observed following the addition of exogenous IAA to the half-strength MS agar medium (Figs 5C, 5E and S11). In *Arabidopsis* mono-associations, IAA-induced RGI was partially suppressed when seedlings were inoculated with strains of *Pseudomonas*, *Variovorax*, *Achromobacter*, *Sphingomonas*, *Curvibacter*, and SB9_*Acinetobacter* (hereafter referred to as RGI-suppressive IAA degraders, while the rest are RGI-non-suppressive IAA degraders) (Fig 5C and 5E). Consistently, a small range of fresh weight enhancement of shoots was observed in these positive mono-associations (Fig 5D). On the other hand, the remaining strains from *Acinetobacter* inhibited primary root elongation, as well as shoot fresh weight (Fig 5C and 5D). No significant effects on root growth or shoot fresh weight were observed in Root154_*Sphingopyxis* treatment. Additionally, in rice mono-associations, compared with axenic control, IAA-induced RGI was partially suppressed when seedlings were inoculated with strains of *Variovorax*, *Achromobacter*, *Curvibacter*, and *Acinetobacter* (S11 Fig).

To explore whether the restoration of RGI by RGI-suppressive IAA degraders is directly related to auxin signaling in plants, the *Arabidopsis* auxin reporter line *DR5::GFP* was treated with IAA and simultaneously inoculated with IAA degraders. Fluorescence of the *DR5::GFP* induced by exogenous IAA remained stable in axenic control at day 1 and day 3 postinoculation. The GFP signal in *DR5::GFP* roots was quenched at day 3 after inoculation with RGI-suppressive IAA degraders. Consistent with the root elongation phenotype, RGI-non-suppressive IAA degraders, including strains from *Sphingopyxis* and *Acinetobacter*, could not quench the

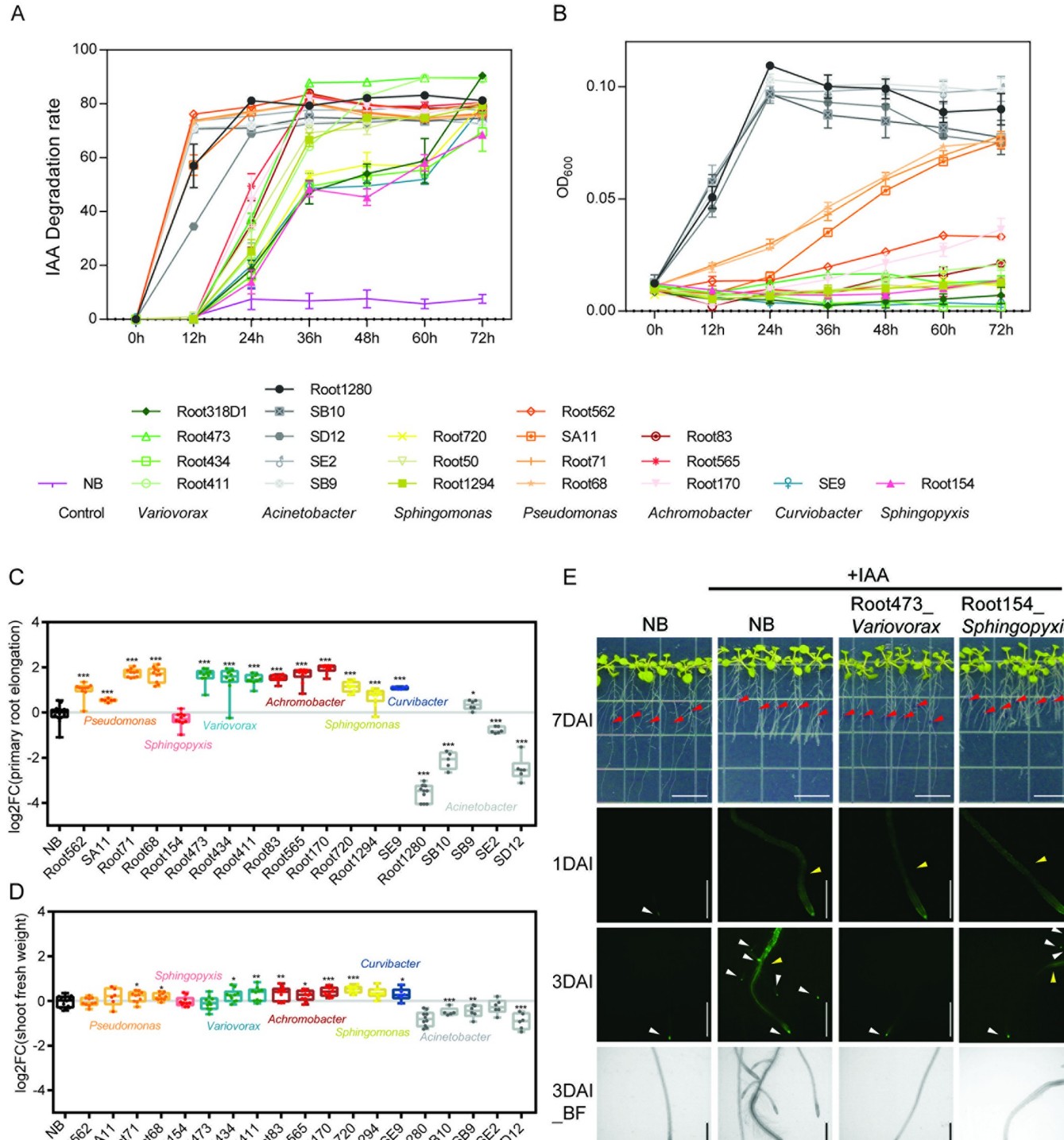

**Fig 5. IAA-degrading strains can utilize IAA and suppress the IAA-induced root growth inhibition.** (A) IAA consumed by IAA degraders in vitro and (B) their growth as measured by $OD_{600}$ in M9 minimal medium supplemented with IAA as the sole carbon source ($n = 3$). (C) Except for strains from genus of *Sphingopyxis* and *Acinetobacter*, IAA-induced root growth inhibition was suppressed by IAA-degrading strains. Boxplot middle line, median $log_2$ fold change of primary root elongation (vs. NB); NB, No bacteria inoculation. Box edges, 25th and 75th percentiles; whiskers, 1.5× the interquartile range. (D) Apart from *Acinetobacter*, strains inoculation have no negative effects on shoot fresh weight. Boxplot middle line, median $log_2$ fold change of shoot fresh weight (vs NB); box edges, 25th and 75th percentiles; whiskers, 1.5× the interquartile range. (E) Images of representative Col-0 seedlings grown axenically (NB) or with IAA-degrading strain inoculation. Upper panels show the representative images of seedlings grown on half strength MS agar plate supplemented with 100 nM IAA at 7 days after inoculation with or without strain. Scale bar = 1.4 cm. Red arrows indicate the original positions of the root tips at the time of bacterial inoculation. The

other panels show the representative primary root images of *DR5::GFP* plants after inoculated with strain for 1 and 3 days. Scale bar = 1 mm. Write arrows show the GFP signal on root tips. Yellow arrows show the GFP signals on root which were induced by exogenous IAA. BF, bright field. Significant differences compared with control group were determined using Student's *t* test: *$P < 0.05$, **$P < 0.01$, ***$P < 0.001$. Numerical values that underlie the data displayed in the panels are in S6 Table. IAA, indole-3-acetic acid.

root fluorescence caused by exogenous IAA (Figs 5E and S12). Interestingly, SB9, the RGI-suppressive IAA degrader from *Acinetobacter*, quenched the fluorescence at day 3 (S12 Fig). To rule out the possibility that the phenomenon observed in RGI-non-suppressive IAA degraders was not caused by failed colonization, colonization of strains was investigated by calculating colony-forming units (CFUs), which were further normalized to root fresh weight. After 7 days of inoculation, all strains successfully colonized roots, and exogenous IAA supplementation had no significant effect on bacterial colonization (S13 and S14 Figs).

To further investigate the potential biological roles of IAA degraders in a natural context, RGI induced by exogenous IAA treatment was replaced with a synthetic bacterial community formed from 110 *Arabidopsis* root isolates (hereafter referred to as Syncom110). Fifteen strains of IAA-degrading bacteria isolated from *Arabidopsis* roots were grouped into different clusters to assess their ability to counteract RGI induced by Syncom110 (S4 Table). The primary root elongation was severely inhibited by inoculation with Symcom110. Interestingly, only clusters containing *Variovorax* strains consistently demonstrated the ability to partially revert the RGI induced by Syncom110 (S15 Fig). These results are consistent with previous report [24].

## Catalogue of potential IAA-degrading bacteria from diverse habitats

IAA degrading bacteria in the rhizosphere play an important role in maintaining auxin homeostasis in roots to ensure normal plant growth and development [25]. Nevertheless, their distribution across various habitats remains limited. To ascertain the distribution of IAA-degrading strains in different habitats, we analyzed a large-scale survey of 11,586 high-quality MAGs (including 750 isolates) collected from mammal gut, aquatic environment, soil, and plants (Fig 6 and S5 Table) [37–45]. The IacA/E and IadD/E were used as the biomarkers to screen the genomes of the collections. We noted that no hits were identified in MAGs collected from cold seeps or animal/human guts, which are reasonable since these habitats are normally hypoxic, while the iac/iad pathways are aerobic (S5 Table). Furthermore, iac or iad-like operon is also absent in the facultative anaerobic environment of the human oral [41]. In contrast, among the 692 MAGs collected from human skin, 5 potential IAA degraders containing iac-like operon were identified [41].

On the other hand, more potential IAA degraders were identified from environment samples, especially plant-associated samples; 0.65% MAGs (2 out of 304) collected from marine samples [41], and 1.63% MAGs (8 out of 492) collected from wastewater samples were identified harboring iad-like operon [39]. In soil samples, 1.31% MAGs (5 out of 382) were identified containing iac- or iad-like operon [39,42]. Within the plant collections, 3.79% MAGs (5 out of 132) affiliated with Sphingomonadales and Burkholderiaceae were identified as containing the iac- or iad-like operon [39]. Furthermore, 7.60% MAGs (60 out of 789, containing 206 isolates) collected from plant shoot [37,38,43] and 12.13% MAGs (66 out of 544, all are isolates) isolated from plant root [37,40] were found to harbor the IAA degradation operon. Overall, there is a consistent increase in the frequency of potential IAA degraders from aquatic to terrestrial environments, from soil to plants, and from plant shoots to roots. Moreover, potential IAA degraders containing iac- or iad-like operon and belonging to Burkholderiales were widely distributed across various habitats (Fig 6).

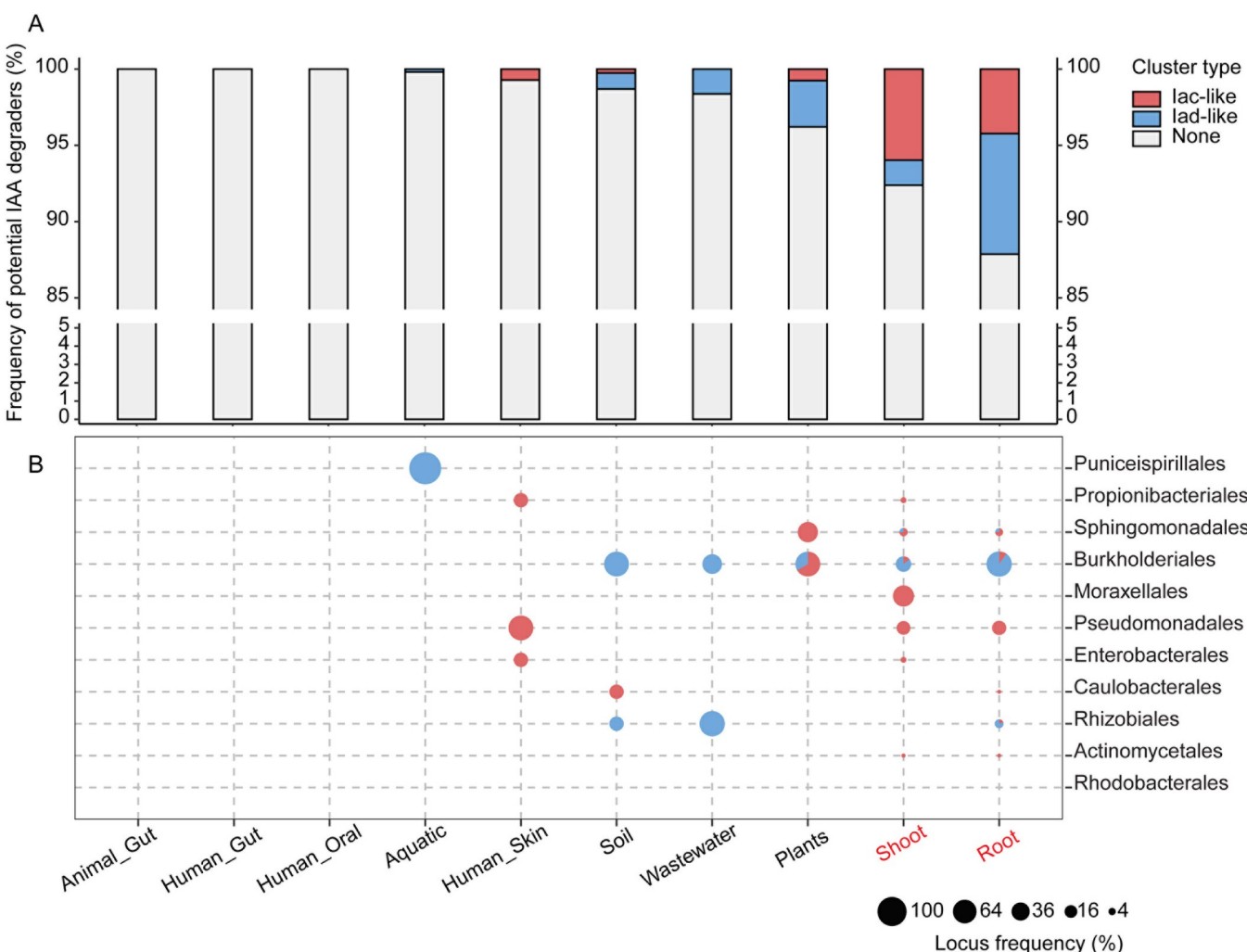

**Fig 6. The distribution of IAA-degrading bacterial strains across various habitats.** (A) The frequency of potential IAA degraders whose genome contains iacA and iacE or iadD and iadE in different habitats. IAA degradation types were labeled with red or blue color. Ggbreak was applied in this analysis [46]. (B) IAA degradation types are varied at bacterial order level. In total, 11,586 high-quality MAGs and isolates were analyzed in this study. Samples containing isolates were highlighted with red color. All MAGs were ≥90% complete, were ≤5% contaminated, and had a quality score (completeness -5 × contamination) of ≥65. Numerical values that underlie the data displayed in the panels are in S6 Table. IAA, indole-3-acetic acid; MAG, metagenome-assembled genome.

## Discussion

In this study, combining genomic analysis and experimental validation, we performed a systematic screening of IAA degraders isolated from *Arabidopsis* and rice root. We found that 21 strains belonging to 7 genera exhibit remarkable IAA degradation activity. In addition to the previously reported *Pseudomonas*, *Achromobacter*, *Variovorax*, *Acinetobacter*, and *Sphingomonas*, strains from *Sphingopyxis* and *Curvibacter* also displayed outstanding IAA-degrading activity. The genomes of all IAA degraders contain iac- or iad-like IAA degradation operon, and these operons were up-regulated by IAA treatment. By integrating protein sequence alignment with protein structural similarity analysis, we revealed that the putative MarR family regulators are structurally conserved within genus. In vitro assays suggested that the screened 21 strains are bona fide IAA degraders. However, only a subset of these strains directly depleted excess IAA to maintain the host-plant root growth. Intriguingly, MAGs analysis shows that IAA-degrading candidates prefer to colonize plant-associated habitats, which seems to suggest

that IAA degraders naturally coexist with IAA producers (here mainly referring to plants and IAA-synthesizing microbes). Our findings reveal potential role of IAA degraders inhabiting in plants and the underlying IAA degradation mechanism.

Comparative genomics studies indicate that the gene clusters of IAA degraders were probably acquired in a natural environment through a horizontal gene transfer pathway, with selective loss, duplication, and rearrangement of IAA-degrading genes during evolution. The gene cluster contains structural genes encoding enzymes responsible for IAA degradation and a set of genes involved in gene expression regulation and compound transportation, and the core components are highly conserved. The structure and arrangement of the iad-like operon in *Variovorax* are highly similar to that in *Achromobacter*, indicating that these 2 genera belonging to Burkholderiales may have obtained the operon from the same ancestor/donor. In contrast, *Curvibacter*, another genus belonging to Burkholderiales, possesses the iac-like operon, suggesting the possibility that IAA degradation gene clusters were obtained through horizontal gene transfer (S6 Fig). *Sphingomonas* and *Sphingopyxis* are closely related and both belong to the Sphingomonadaceae. A complete iac-like operon is uniquely present in the genomes of strains from *Sphingomonas*. Moreover, IAA-degrading gene cluster mining analysis showed that 2 additionally fragmentary iad-like operons exist on *Sphingomonas* genomes, and one of them has the same gene arrangement to the *Sphingopyxis* (S5 Fig), suggesting the evolutionary homology of these 2 genera. Additionally, although *Acinetobacter* and *Curvibacter* belong to different families, both genera have iac-like operons and the operons have the similar gene arrangement, suggesting that they may obtained the gene cluster from a closer donor (Fig 2).

IacR and IadR are an essential component of the IAA-degrading gene cluster, and functional studies suggest that it normally serves as an expression suppressor of the operon. Recently, the protein crystal structure of IadR was resolved, and it was confirmed that IadR binds to the upstream DNA sequence of *iadA* to inhibit iad locus expression in *Variovorax paradoxus* CL14 [24]. The presence of IAA results in IadR being released from the DNA binding site, further disinhibits the expression of the iad operon. The MarR regulators have highly conserved protein structures despite having highly differentiated protein sequences (Figs 3, S7 and S8). This might explain why different MarR regulators appear to perform similar functions in regulating operon expression. Similar to *Paraburkholderia phytofirmans* PsJN, LuxR and IacS uniquely exist in the iac-like operons of 4 strains belonging to *Pseudomonas*, suggesting that a putative two-component regulatory system independently evolved or obtained in this genus (Fig 2) [20].

Plant growth requires multifaceted regulation, including but not limited to IAA production and degradation by plant, as well as auxin regulation by plant-associated microbiota [25]. It is estimated that 80% of rhizosphere commensal isolates possess the capability of producing IAA [15]. While, based on our results, only 11.48% of the isolates (21 out of 183 in this study) and 7.84% (131 out of 1,465) of the plant-associated MAGs (S5 Table) exhibit potential IAA-degrading capability. The results of bacterial inoculation experiments suggest a potential role for IAA degraders in influencing IAA homeostasis in plants, potentially contributing to the maintenance of root growth. However, not all IAA-degrading strains have the ability to reverse the severe RGI induced by excess IAA or a synthetic bacterial community. Taking *Pseudomonas* as an example, the addition of IAA to the MS medium may have triggered the transcription of the *iac*-like operon, enabling IAA degradation and further partially alleviating RGI phenotype in *Arabidopsis* caused by exogenous IAA. However, in a more complex microbial community, introduced bacterial members may engage in direct antagonistic interactions with *Pseudomonas*, potentially preventing the transcriptional activation of *iac*-like operon due to a trade-off in the competitive dynamics. Additionally, *Pseudomonas* is also a notable producer of IAA. Studies have reported that IAA synthesis is a crucial mechanism enabling

commensal bacteria to resist plant-generated reactive oxygen species (ROS) and establish successful colonization [14]. In the presence of competitors, *Pseudomonas* may activate its IAA synthesis pathway to enhance colonization on plant roots. This hypothesis based on the current research findings, though the precise molecular mechanisms remain to be further investigated.

Notably, inoculation with *Variovorax* not only restored root growth in *Arabidopsis* and rice induced by exogenous IAA, but also partially alleviated RGI caused by Synthetic bacterial community inoculation. This finding is consistent with recent study, and further validate the importance of *Variovorax* in plant root morphogenesis [24,25]. *Variovorax* may assist plants in regulating IAA concentrations through its IAA degradation capability, thereby preventing RGI caused by excessive IAA in rhizosphere produced by other microbes. Additionally, compared to the axenic control, inoculation with *Variovorax* alone can promote primary root growth (S15 Fig). Co-inoculation with Syncom110 showed partial reversal of RGI, which may also be attributed to the general root growth-promoting effect of *Variovorax*. Moreover, members from this genus may be involved in the regulation of IAA signaling or interact with other plant hormones, such as cytokinins, which warrants further investigation.

On the other hand, our mono-association results showed that *Acinetobacter* has negative effects on *Arabidopsis* growth (Figs 5 and S12), which is inconsistent with recent reports that *Acinetobacter* can act as a plant growth-promoting rhizobacteria (PGPR) [47,48]. In addition to being a strong IAA-degrading bacterium, *Acinetobacter* is also one of many that produce IAA and can thus sabotage plant physiology by adding to the endogenous IAA pool in plants [49]. This dual functionality may reflect the bacteria' diversity or condition-dependent behavior in different habitat. Therefore, the biological role of *Acinetobacter* and its underlying mechanisms in plants require further investigation, particularly its specific position within the IAA regulatory network and its interactions with other plant hormones. Attention should be given to whether *Acinetobacter* exhibits different IAA metabolism strategies across various plant hosts or environmental conditions, in order to better understand its overall impact on plant physiology.

Lastly, hormones may play a crucial role in mediating the interactions between hosts and microbes. For instance, *Mycobacterium neoaurum* possessing the capability to degrade testosterone was isolated from the fecal samples of testosterone-deficient patients with impression. Further experiments revealed the potential association between human gut microbes expressing 3β-HSD and depressive symptoms resulting from testosterone degradation [50]. Beyond IAA, other plant hormones have been reported to be synthesized or metabolized by microbes. For example, rhizobacteria such as *Rhodococcus* sp. P1Y and *Novosphingobium* sp. P6W have been reported to utilize abscisic acid (ABA), consequently stimulating plant growth through an ABA-dependent mechanism [51]. We envision that investigations on microbial metabolism of host hormones may offer novel insights into the understanding of homeostasis, host physiology, and the development of diseases.

## Materials and methods

### Plant materials and bacterial strains

Seeds of *Arabidopsis* (ecotype Columbia, Col-0) and rice (cv. Nipponbare) were obtained from laboratory stock. The *Arabidopsis* auxin reporter transgenic line *DR5::GFP* was kindly provided by Prof. Xugang Li (Shandong Agriculture University).

The 131 bacterial commensals isolated from *Arabidopsis* roots or soil were gifts from Prof. Paul Schulze-Lefert (Max Planck Institute for Plant Breeding Research, Cologne, Germany) [37]. Detailed information on individual strains can be found at At-RSPHERE (http://www.at-sphere.com/).

The 52 rice root-associated bacterial isolates analyzed in this study were retrieved from our laboratory stock. In detail, rice root samples contain bulk soil were collected in field and immediately delivered with ice to our lab. After removed the bulk soil, 10 g root samples contain rhizosphere soil were washed multiple times with sterilized water until there is no obvious soil on root surface. The washings were mixed as the rhizosphere sample. Rice roots were then grind with 10 ml 1 × PBS buffer in a sterilized mortar and filtered with sterilized gauze. Both samples were then spread on the surface of $^1/_5$ tryptic soy broth (TSB, 6 g/L, Sigma-Aldrich, United States of America) agar plates with series dilution. After incubation for 3 days at 25°C, single colonies were randomly picked from the plates for twice purification. A total of 207 isolates were identified at the species level by sequencing 16S rRNA gene with the primers 27F (AGAGTTTGATCCTGGCTCAG) and 1492R (GGTTACCTTGTTACGACTT). For whole genome sequencing, the genome DNA of selected 72 strains were individually extracted using FastDNA Spin Kit for Soil (MP Biomedicals, USA). Library preparation was performed using the Hieff NGS OnePot II DNA Library Prep Kit (Yeasen, China) for Illumina with 50 ng DNA per sample. The draft genomes were generated with the HiSeq Xten platform (Illumine, USA). Quality control of the raw reads were filtered with fastp [52], followed with genome assembly through Unicycler [53]. CheckM was used to estimate the quality of each genome, including the numbers and N50 of the contigs, the contamination, and the completeness [54]. Prokka was used to annotate the function of all assembled genomes [55]. Taxonomy annotation of the isolates was performed using the Genome Taxonomy Database Toolkit (GTDB-Tk) [56] with reference to GTDB release 207 [57]. Isolates were assigned at the species level if the ANI to the closest GTDB-Tk representative genome was ≥95% and the aligned fraction was ≥60%. General taxonomical information of these 183 strains is listed in S1 Table.

## Construction and modification of phylogenetic tree

The phylogenetic trees based on whole genome sequences of strains were constructed using phylophlan [58]. While phylogenetic trees based on protein sequences were constructed using MUSCLE [59] for multiple sequence alignment and MEGA [60] for tree construction. Specifically, Figs 1 and S3 was constructed using the WGS of 183 strains; Fig 2A was constructed using the amino acid sequences of IAA metabolism gene clusters obtained from 21 IAA-degraders; Fig 2B was constructed using the amino acid sequences of 21 MarR proteins discovered from the 17 IAA-degraders and 6 related templates. The generated phylogenetic trees were further visually modified using iTOL [61].

## Identification of potential IAA-degraders

Prokka [55] was used for functional annotation of isolates and MAGs derived from different environments, meeting the criteria of genome completeness (≥90%) and contamination (≤5%). Diamond [62] was employed for sequence alignment of annotated genomes, using previously reported IacA and IacE or IadD and IadE sequences as templates. The alignment filtering threshold was set at sequence similarity (Identity) >50% and sequence coverage (Coverage) >60%. If both gene combinations were simultaneously identified in a genome and determined to be located in the same gene cluster (*iadD* adjacent to *iadE*; *iacA* with a distance less than 7 Coding Sequences (CDS) from *iacE*), the strain was considered to possess IAA degradation capability.

## Bacterial culture and screening of IAA degradation

Individual bacteria from glycerol stock were incubated on half-strength TSB (15 g/L) agar plates at 25°C for 5 days. A single colony of each strain was then cultured in $^1/_2$ strength TSB

liquid medium at 25°C with 400 rpm shaking. When the cell culture reached the exponential growth phase, the optical density of the culture was measured at 600 nm ($OD_{600}$) using a Synergy H1 microplate reader (BioTek, USA).

The bacterial culture was washed once with 1×PBS and then added to 1 ml of $^1/_2$ strength TSB medium supplemented with or without 0.4 mM IAA (dissolved in ethanol) (Sigma-Aldrich, USA) to achieve a final $OD_{600}$ of 0.05. The control group had the same volume of ethanol added to the medium. After 72 hours of incubation at 25°C with 400 rpm shaking, the IAA content of each sample was measured using the colorimetric assay [30]. Briefly, 120 μl of Salkowski reagent (a muxture of 0.5 M ferric chloride and 35% sulfuric acid) was mixed with 60 μl of culture supernatant. The residual IAA in the medium reacts with $Fe^{3+}$ to form a pink compound, which exhibits an absorption peak at 530 nm. After 30 minutes of incubation in the dark, the absorbance was measured using a Synergy H1 microplate reader. The remaining IAA contents in the medium were then calculated using IAA standard curves. The bacterial degradation rate was further calculated as the IAA consumed divided by the initial IAA content in the culture.

## Validation of IAA degradation by LC-MS analysis

To validate the IAA degradation results of the colorimetric assay, cell cultures of 21 strains were further analyzed by LC-MS [30]. After 72 hours of incubation with IAA, the supernatant from each bacterial culture was collected to detect the IAA compound in the medium. In detail, the residual IAA of the bacterial culture was extracted once with ethyl acetate, followed by volatilization, and the residue was further dissolved in methanol. After filtration, a 2 μl sample was separated using a C18 column (Infinity Lab Poroshell 120 EC-C18, 2.1 × 50 mm, 2.7 μm; Agilent, USA) connected to the Agilent 6470B triple quadrupole LC/MS (Agilent, USA). Solvent A (water supplemented with 0.1% formic acid) and solvent B (methanol) were used as mobile phases at a flow rate of 0.4 ml/min under a gradient elution: 0 to 0.3 min, 40% B; 0.3 to 4 min, 95% B; 4 to 4.5 min, 95% B; 4.5 to 4.55 min, 40% B; 4.55 to 5.5 min, 40% B. The quantification of IAA extracted from culture and IAA standards was performed using the positive-ion multiple reaction monitoring (MRM) method.

## Growth experiment with IAA as the sole carbon source

Selected IAA degraders were individually pre-cultured at 25°C with shaking at 200 rpm in 2 ml of $^1/_2$ TSB medium. Bacterial cells were harvested by centrifugation, and the pellet was washed twice with 1×PBS. IAA degraders were then individually cultured at 25°C with shaking at 200 rpm in 1.5 ml of M9 minimal salts medium (Sigma-Aldrich, USA), supplemented with 2 mM $MgSO_4$ and 0.1 mM $CaCl_2$. A 0.4 mM IAA solution was added to the culture as the sole carbon source. The control group had the same volume of ethanol (0.05% v/v) added to the medium. The initial $OD_{600}$ for the experiment was set at 0.02. $OD_{600}$ and IAA concentrations of the cultures were measured at 7 time points: 0 h, 12 h, 24 h, 36 h, 48h, 60 h, and 72 h.

## RNA-seq and data analysis

Root71_*Pseudomonas*, Root170_*Achromobacter*, Root473_*Variovorax*, SE9_*Curvibacter*, and Root154_*Sphingopyxis* were grown in M9 medium supplemented with 1 mM IAA or 1.712 mM glucose (same volume of ethanol was added in glucose treatments). The initial $OD_{600}$ of the cultures was 0.05, and they were incubated at 25°C with 400 rpm shaking. Cells from Root71_*Pseudomonas*, Root170_*Achromobacter*, Root473_*Variovorax*, SE9_*Curvibacter*, and Root154_*Sphingopyxis* were collected for RNA-seq at 14 h, 16 h, 14 h, 48 h, and 20 h,

respectively. Bacterial pellets were collected by centrifuging the culture at 12,000 rpm for 10 minutes at 4°C. The pellets were stored at −80°C until RNA extraction.

Further experiments including RNA extraction, library preparation, and sequencing were performed at Magigene Co. Ltd using the Nova Seq6000 platform (Illumina, USA). In detail, bacterial RNA samples were extracted using the Trizol followed with quality control by Thermo NanoDrop One and Agilent 4200 Tape Station. Epicentre Ribo-Zero rRNA Removal Kit was using to remove Ribosome RNA in the samples. Library preparation was performed using the NEBNext Ultra Directional RNA Library Prep Kit for Illumina (New England Bio-labs; USA) with 1 μg total RNA per sample. The sequence data were processed with a custom RNA-seq pipeline from Low quality reads were filtered with fastp (v0.23.4) [63]. The clean reads were then mapped to the draft genome of corresponding strain with STAR (v2.7.10a) [64]. Gene quantification was subsequently done using SAMtools (1.18) [65] and RSEM (1.3.3) [66]. DEGs were identified using DESeq2 (v1.36.0) [67] with $\log_2^{\mathrm{FoldChange}} \geq 1$ or $\leq -1$ and adjusted $p$-value $< 0.05$ as cutoffs. Three biological replicates of each sample were used to perform the transcriptome analyses.

## Plant experiments

Strains selected for in planta assays were pre-cultured in $^1/_2$ strength TSB at 25°C, 200 rpm for 2 to 3 days. On the day of inoculation, the bacterial culture was subcultured at a 1:1 ratio for an additional 5 hours. For mono-association of individual IAA-degrader inoculation, a 500 μl aliquot of bacterial culture was centrifuged at $6,500 \times g$ for 2 minutes. After washing twice with 10 mM $MgCl_2$, the bacterial pellets were resuspended in 10 mM $MgCl_2$ and adjusted to an $OD_{600}$ of 0.01 (0.1 for rice seedling inoculation). A 100 μl aliquot of bacterial suspension was then spread on half-strength Murashige and Skoog (MS) (Sigma-Aldrich, USA) plates supplemented with or without 100 nM IAA (10 μm IAA for rice seedling inoculation).

For 110-member (Syncom110) synthetic bacterial community inoculation, the $OD_{600}$ values of 110 strains (S4 Table) were measured, and 50 μl bacterial culture of each strain was pooled together. The mixture was then centrifuged at $6,500 \times g$ for 2 minutes, and the pellet was washed twice with 10 mM $MgCl_2$. Prepared the Syncom110 inoculum by resuspending the pellet in 10 mM $MgCl_2$ and diluting the $OD_{600}$ to 0.2. Bacterial cultures of 15 IAA-degraders isolated from *Arabidopsis* root were collected individually by centrifugation and washed twice with 10 mM $MgCl_2$. The bacterial pellets were resuspended in 10 mM $MgCl_2$ and adjusted to an $OD_{600}$ of 0.01. Preparation of the different groups of IAA degraders (S4 Table) by mixing each strains at equal proportions; 50 μl of Syncom110 inoculum ($OD_{600} = 0.2$) combined with 50 μl 10 mM $MgCl_2$ (as NB control) or 50 μl different groups of IAA-degrader ($OD_{600} = 0.01$) were then spread on $^1/_2$ strength MS agar plate.

*Arabidopsis* seeds were surface-sterilized with 75% ethanol for 1 minute, 20% bleach for 15 minutes, and rinsed 5 times with sterile distilled water. Seeds were sown evenly on $^1/_2$ strength MS plates with 0.5% agar and 3% sucrose. After 2 days of stratification at 4°C in the dark, seeds were vertically grown in a growth chamber under a 16-h dark/8-h light regime at 22°C for 7 days. Ten seedlings were transferred to the prepared half-strength MS plates containing IAA and IAA-degrading bacteria. The initial position of the root tip was labeled, and after an additional 7 days of growth, the final position of the root tip was labeled again. Pictures of the plates were captured with a camera (Nikon, Japan), and the elongation of the primary root was measured using ImageJ [68].

For rice seedling inoculation, the rice seed was dehulled, and half of the endosperm was removed to reduce contamination. Seeds were then surface-sterilized with 75% ethanol for 1 minute, 100% bleach for 5 min, and rinsed 5 times with sterile distilled water, and sown on

half-strength MS plates. Seeds were vertically grown in a growth chamber under a 12-h dark/ 12-h light regime at 27°C for 2 days. Five to 6 seedlings were transferred to the prepared half-strength MS plates containing IAA and IAA-degrading bacteria. The initial position of the root tip was labeled, and after an additional 3 days of growth, the final position of the root tip was labeled again. Pictures of the plates were captured with a camera (Nikon, Japan), and the elongation of the primary root was measured using ImageJ.

### Fluorescence microscopy

GFP fluorescence in the roots of *DR5::GFP* transgenic lines was visualized using a Ti2-E fluorescence microscope (Nikon, Japan) at 1 and 3 days after inoculation, respectively. The experiment was performed in 2 independent replicates.

### Measurement of bacteria root colonization

CFUs were counted as previously described with minor modifications [69]. Briefly, after 7 days of inoculation, roots were separated from shoots using a sterile scalpel, taking care to avoid contamination between different bacterial treatments. Two roots were placed in pre-weighed sterile tubes containing metal beads, and the tubes were weighed again to obtain the fresh weight of the roots. Root samples were then homogenized using a TissueLyzer (Shanghai Cebo, China) at 30 Hz for 30 seconds. A 500 μl aliquot of 1×PBS buffer was added to the tube, and the samples were serially diluted in a sterile 96-well plate. A 5 μl liquid sample was placed on the far left side of the $^1/_2$ TSB agar plate. The plate was slowly tilted to the right to allow the liquid to flow evenly. Once the liquid reached a certain point, the plate was slowly tilted to the left, causing the remaining liquid to move back towards the left side. Finally, the plates were placed at 25°C for 2 days until single colonies appeared. The colonization ability of each strain on the root was calculated according to the CFU count.

### Supporting information

**S1 Fig. Aerobic IAA degradation pathways identified in microbes.** Blue arrows showed the iad-pathway, the typical examples are *Bradyrhizobium japonicum* [23] and *Variovorax paradoxus* [24]. Red arrows showed the iac-pathway, the typical examples are *Pseudomonas putida*, *Paraburkholderia phytofirmans*, and *Caballeronia glathei* [18,20,22]. Black arrows showed pathways were not well classified so far. In 1961, Tsubokura and colleagues reported that IAA was biotransformed to 2-formaminobenzoylacetate and fruther decomposed into anthranilate by a bacterium isolated from air [70]. It is reported that *Lysinibacillus xylanilyticus* strain MA transformed tryptophan to IAA and then decarboxylated to produce 3-methylindole (skatole) [71], and the downstream pathways of skatole catabolism were firstly identified in *Pseudomonas* (Migula) in 1958 [72].
(PDF)

**S2 Fig. The iac and iad gene clusters from previous studies [16,18,20–24,27,49] were used as template sequences for comparative genomics analysis.**
(PDF)

**S3 Fig. The IAA catabolism gene clusters were annotated in the bacterial strains.** The whole genome sequence-based phylogenetic tree of the 183 strains was generated with phylophlan and visualized with iTOL. Genes annotated as iac-like or iad-like operons (with over 40% identity and 60% coverage compared to template amino acid sequences) were labeled with triangles.
(PDF)

**S4 Fig. LC-MS validation of IAA degradation in 21 strains.** Retention time of IAA in samples are consistent with commercial standard.
(PDF)

**S5 Fig. Two sets of fragmentary iad gene clusters were found in the genomes of *Sphingomonas* strain Root1294, Root720, Root50, and *Sphingopyxis* strain Root154.**
(PDF)

**S6 Fig. The phylogenetic tree based on concatenated IacA and IacE or IadD and IadE amino acid sequences vs. the phylogenetic tree based on whole genome sequences (WGS) of the strains.**
(PDF)

**S7 Fig. Sequence alignment of the 21 MarR proteins identified in 17 strains of this study and the 6 reference MarR proteins.** Six MarR templates used in this analysis (labeled in red) are 7L1I *Acinetobacter baumannii*, 7KUA *Pseudomonas putida*, 7L19 *Enterobacter soli* ATCC BAA-2102, 7KYM *Bradyrhizobium japonicum*, 3CDH *Ruegeria pomeroyi* DSS-3, and 7KFO *Variovorax paradoxus* CL14.
(PDF)

**S8 Fig. Predicted protein structures of the MarR proteins identified in IAA-degrading strains.** All the structures were predicted by AlphaFold2.
(PDF)

**S9 Fig. KEGG level1 pathways enrichment for the differentially expressed genes under IAA treatment.** Numerical values that underlie the data displayed in the panels are in S6 Table.
(PDF)

**S10 Fig. IAA-degrading strains can utilize IAA as carbon source.** Growth of individual IAA-degrading strains was measured at $OD_{600}$ in M9 minimal medium supplemented with IAA (dissolved in ethanol) or M9 minimal medium supplemented with 0.05% ethanol as the sole carbon source ($n = 3$). Significant differences compared between M9+IAA and M9+0.05% ethanol were determined using Student's $t$ test: $^*P < 0.05$, $^{**}P < 0.01$, $^{***}P < 0.001$. Numerical values that underlie the data displayed in the growth curves are in S6 Table.
(PDF)

**S11 Fig. IAA-degrading bacterial strains restore the RGI-induced by exogenous IAA in rice seedlings.** (A) Exogenous IAA induced RGI was suppressed by some of the IAA-degrading strains. Boxplot middle line, median $\log_2$ fold change of primary root elongation (vs. NB); box edges, 25th and 75th percentiles; whiskers, from the minimum to the maximum value. Each point represents a biological replicate, with 4 to 6 replicates were used in this experiment. (B) The images of rice seedlings grown on half strength MS agar plate supplemented with 10 μm IAA at 3 days after inoculation with or without bacteria. Red arrows indicate the original positions of the root tips at the time of bacterial inoculation. NB, No bacterial inoculation. Significant differences compared with NB group were determined using Student's $t$ test: $^*P < 0.05$, $^{**}P < 0.01$, $^{***}P < 0.001$. Numerical values that underlie the data displayed in the plot are in S6 Table.
(PDF)

**S12 Fig. IAA-degrading bacterial strains restore RGI induced by exogenous IAA.** The upper panels show the images of seedlings grown on $^1/_2$ strength MS agar plate supplemented

with 100 nM IAA, taken 7 days after inoculation with or without bacteria. Scale Bar = 1.4 cm. NB, No bacterial inoculation. Red arrows indicate the original positions of the root tips at the time of bacterial inoculation. The other panels show images of the primary roots of *DR5::GFP* plants following bacterial inoculation at 1 and 3 days postinoculation. Scale Bar = 1 mm. Write arrow shows the GFP signals on root tip. Yellow arrow shows the GFP signals on root which was induced by exogenous IAA. BF, bright field.
(PDF)

**S13 Fig. Colonization by individual IAA-degrading bacterial strains on *Arabidopsis* seedling root at 7 days postinoculation.** The log-transformed CFUs of the IAA-degrading bacteria normalized to corresponding root weight on half-strength MS agar (open bar) and MS agar supplement with 100 nM IAA (solid bar) ($n$ = 3). Numerical values that underlie the data displayed in the graph are in S6 Table.
(PDF)

**S14 Fig. Correlation between primary root elongation and colonization of IAA degraders after 7 days inoculation.** Each blue dot shows mean of 3 to 4 biological replicates for 18 IAA degraders. Numerical values that underlie the data displayed in panel are in S6 Table.
(PDF)

**S15 Fig. IAA-degrading bacterial strains restore RGI caused by Syncom110 in *Arabidopsis* seedlings.** (A) The complex synthetic bacterial community (comprising 110 bacterial strains, see S4 Table) induced RGI was partially alleviated by introducing IAA-degrading strains. Box-plot middle line, median primary root elongation; box edges, 25th and 75th percentiles; whiskers, from the minimum to the maximum value. Each point represents a biological replicate, with 4 to 6 replicates were used in this experiment. (B) The images of *Arabidopsis* seedlings grown on half-strength MS agar plates inoculated with or without bacteria at day 7. Red arrows indicate the original positions of the root tips at the time of bacterial inoculation. NB indicates no bacterial inoculation. Significant differences compared with NB group were determined using Student's $t$ test: *$P < 0.05$, **$P < 0.01$, ***$P < 0.001$. Numerical values that underlie the data displayed in the panels are in S6 Table.
(PDF)

**S1 Table. General taxonomic information for the 183 strains analyzed in this study.**
(XLSX)

**S2 Table. The amino acid sequences of the candidate Iac or Iad-like operons were compared to the templates.**
(XLSX)

**S3 Table. IAA-induced differentially expressed genes (DEGs) were identified among the five studied strains.**
(XLSX)

**S4 Table. Information for the Syncom110 and groups of IAA-degrading strains.**
(XLSX)

**S5 Table. High-quality metagenome-assembled genomes (MAGs) were collected for the analysis of iac and iad-like operons.**
(XLSX)

**S6 Table. Data details for generating graphs in this study.**
(XLSX)

## Acknowledgments

We gratefully acknowledge Prof. Paul Schulze-Lefert for sharing *Arabidopsis* associated microbes. We gratefully acknowledge Prof. Xugang Li for providing *Arabidopsis* auxin reporter transgenic line *DR5*::*GFP*.

## Author Contributions

**Investigation:** Lanxiang Wang, Yue Liu, Haoran Ni, Lei Dai.

**Methodology:** Lanxiang Wang, Yue Liu, Haoran Ni, Haimei Shi, Weixin Liao.

**Project administration:** Lei Dai.

**Resources:** Moxian Chen, Lei Dai.

**Software:** Haoran Ni, Wenlong Zuo, Hongbin Liu.

**Supervision:** Moxian Chen, Lei Dai.

**Validation:** Lanxiang Wang, Yue Liu, Jiajia Chen.

**Writing – original draft:** Lanxiang Wang.

**Writing – review & editing:** Lanxiang Wang, Yang Bai, Hong Yue, Ancheng Huang, Jonathan Friedman, Tong Si, Yinggao Liu, Moxian Chen, Lei Dai.

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
