## [Editor Report · Decision Letter 0]

17 Jul 2024

***HAVE YOU ASSIGNED THE AE? IF NOT CANCEL THIS DECISION AND GO BACK!***

***REMEMBER TO CLOSE ALL REDUNDANT AE DISCUSSIONS***

Dear Dr Dai, 

Thank you for submitting your manuscript entitled "Degradation of indole-3-acetic acid by plant-associated bacteria" for consideration as a Research Article by PLOS Biology addressing the concerns highlighted in the previous submission.

Your revised manuscript has now been evaluated by the PLOS Biology editorial staff, as well as the previous academic editor with relevant expertise, and I am writing to let you know that we would like to send your submission back to the previous panel of reviewers.

Once your full submission is complete, your paper will undergo a series of checks in preparation for peer review. After your manuscript has passed the checks it will be sent out for review. To provide the metadata for your submission, please Login to Editorial Manager (https://www.editorialmanager.com/pbiology) within two working days, i.e. by Jul 19 2024 11:59PM.

Kind regards,

Melissa

Melissa Vazquez Hernandez, Ph.D.

Associate Editor

PLOS Biology

---

## [Decision Letter · Decision Letter 1]

8 Sep 2024

Dear Dr Dai,

Thank you for your patience while we considered your new submission with the revised manuscript "Degradation of indole-3-acetic acid by plant-associated bacteria" for consideration as a Research Article at PLOS Biology. Your revised study has now been evaluated by the PLOS Biology editors, the Academic Editor and the original reviewers. 

As you will see in the reports, all reviewers recognize the improvements made in addressing their previous concerns. However, Reviewers #2 and #3 still have a few remaining points that need attention. Reviewer #2 remains unconvinced about the biological relevance and suggests strengthening the evidence for the role of IAA degradation in root growth inhibition (RGI) suppression. Reviewer #3 has minor comments concerning protein structure similarity, a missing control for bacterial growth, and recommends expanding the discussion.

IMPORTANT: While the additional experiments suggested by Reviewer #2 would indeed strengthen the study's conclusions, we leave it to your discretion to conduct them. As the Academic Editor notes, "It's okay to ask the authors to respond to the reviewers, but I think modifying the claims and acknowledging the correlative nature of this result would be sufficient." Therefore, if no further experiments are planned, please revise the claims to reflect the correlative relationship between SynCom-mediated root growth inhibition and its reversal.

**IMPORTANT - SUBMITTING YOUR REVISION**

*Resubmission Checklist*

*Published Peer Review*

*PLOS Data Policy*

*Blot and Gel Data Policy*

Sincerely,

Melissa

Melissa Vazquez Hernandez, Ph.D.

Associate Editor

PLOS Biology

REVIEWERS' COMMENTS

Reviewer #1: 

All of my comments have been addressed.

Reviewer #2: 

With their new submission, Wang and colleagues have carefully addressed the reviewers' comments on the previously submitted manuscript and provided detailed responses to the questions. Additional in planta experiments and new computational analyses were performed that improve clarity and comprehensiveness of the study and relevance of the findings. While I appreciate a lot the effort, and the paper thoroughly dissects the potential of IAA degradation in root-associated bacteria, my major concern remains that there is still an overinterpretation of the results with respect to the biological role of bacterial IAA degraders in association with a plant host. 

In a new experiment, Arabidopsis plants were germinated and transferred to agar plates containing a diverse synthetic bacterial community of 110 isolates, plus either no other bacteria, or 15 IAA degraders, or sets of members of those 15. As a control, agar plates without the Syncom were used. There is root growth inhibition (RGI), i.e., decreased primary root length elongation, when the Syncom is present. The authors interpret this as a result of IAA produced by the Syncom, which is a very speculative claim. Nevertheless, the addition of some of the degraders causes suppression of RGI, which is quite interesting. This suppression by one or all of the 15 members of IAA degraders could be a consequence of several IAA-independent scenarios. Microbes antagonizing the ones that cause RGI, general root growth promotion by individual strains, etc. If the authors could show that indeed IAA concentration on the root/in the system in planta decreases, the evidence would be much stronger. Another option could be to generate a bacterial mutant that lacks the capability of IAA consumption or degradation and test if RGI suppression is abolished. These may be difficult experiments to establish. However, as long as direct evidence for the relevance of IAA degradation on the plant root is missing, it is impossible to conclude that RGI suppression is the biological role of IAA degraders. It is then crucial to be very careful in the phrasing and refrain from posing strong claims like "key role" and "reveal the biological role of IAA homeostasis". 

Regarding the presentation of the data, I find the photos of the plants on agar plates a bit confusing when directly compared with the root length elongation data. The elongation is of course not visible in the plant photos. It may make sense to make this clearer either by saying explicitly in the figure caption that, although absolute root length of the plants may be similar between conditions, this is a consequence of differential root length elongation, or by indicating on the photos where the initial root tip was after transfer to the plates. 

Minor comments underlining the advice to thoroughly proof-read the manuscript with respect to grammar, phrasing, and accuracy:

Salkowski method still appears in the text (e.g., line 153) and in the figures (e.g., Fig. 1). 

Revise grammar in Figure S11. Also, not clear if NB-IAA means with or without IAA. 

Line 401-403: This sentence does not seem to be complete. 

Line 422: suggesting instead of suggested

Line 521: What does half-strength TSB mean? How much g/L?

Line 533: Please explain in a bit more detail what the Salkowski method is and what it visualizes.

Line 575: Illumina

Reviewer #3: 

This revised version of the manuscript successfully addresses most of the comments from my original review and is a clear improvement over the previous version. The addition of relevant new in planta experiments and results has strengthened the study, making it more comprehensive. I enjoyed reading the manuscript, and I believe the findings presented are valuable to the field.

The primary contribution of this study is its demonstration that bacteria capable of degrading auxin are relatively common within the plant microbiome. In this revision, the authors have responded to critiques by incorporating experiments that explore the in planta significance of auxin degradation by the root microbiome. Notably, they show that certain IAA-degrading strains can partially revert the root growth inhibition (RGI) phenotype induced by a synthetic community (Syncom110). Although the exact biological role of auxin degradation remains unclear, the study suggests that some of these bacteria may use indole-3-acetic acid (IAA) as a carbon source and could potentially influence root development. Thus, the study is primarily incremental and aligns well with and supports the conclusions published by Finkel et al. in 2020.

A few additional comments are provided below.

1. I am still confused about the TM-score shown in Fig. 2B. If this is a measure of similarity between two protein structures, I would expect that 'A vs. B' would yield the same score as 'B vs. A'. However, this is not the case in this figure. As I am not an expert in protein structures, I cannot fully assess whether the analysis was done correctly. This lack of understanding raises a question: What is the actual result in this figure? What do we learn from it, given that the score varies depending on the direction of the comparison? The authors claim that IacR and IadR show high similarity to known MarR proteins, a conclusion I agree with based on Supplementary Figures 7 and 8. However, the presentation in Fig. 2B is confusing. What is the TM-score threshold for "high similarity"?

2. Figure 4B shows the growth of the bacterium in M9 medium supplemented with IAA as the sole carbon source. The methods section indicates that M9 medium with ethanol (the IAA solvent) was used as a control. However, the result of the control condition is not clearly shown or used in the figure. Bacterial growth in medium with and without IAA should be presented side by side to better support the conclusion that some of these strains indeed use auxin as the carbon source.

3. In Supplementary Figure 11, the first treatment is labeled "NB-IAA," which could be misinterpreted as "No bacteria and addition of IAA." The authors could prevent this misunderstanding by using a more explicit label, such as "NB without IAA."

4. The discussion on the biological role of IAA degradation by plant-associated microbes is somewhat superficial. For instance, the authors mention that IAA-degrading bacteria have potential roles in regulating plant development. However, only one genus (Variovorax) could revert the root growth inhibition (RGI) phenotype induced by SynCom110 in a "natural" context. If only Variovorax can revert the RGI phenotype, what could be the role, if any, of the other IAA-degrading strains in plant development? More effectively highlighting and discussing the possibility that auxin degradation by the plant microbiome serves purposes yet to be discovered could set this paper apart from previous work on this topic.

5. Line 269: Replace "assay" with "assays."

6. Line 285: Replace "although" with "though."

7. Line 294: Replace "additional" with the actual number of days in "After additional days of growth, primary root elongation was measured."

---

## [Editor Report · Decision Letter 2]

23 Oct 2024

Dear Dr Dai,

Thank you for your patience while we considered your revised manuscript entitled "Degradation of indole-3-acetic acid by plant-associated bacteria" for publication as a Research Article at PLOS Biology. This revised version of your manuscript has been evaluated by the PLOS Biology editors and by the Academic Editor.

Based on our Academic Editor's assessment of your revision, we are likely to accept this manuscript for publication, provided you satisfactorily address the data and other policy-related requests stated below.

In addition, we would like you to consider a suggestion to improve the title:

"Systematic characterization of plant-associated bacteria that can degrade the plant hormone indole-3-acetic acid produced by Arabidopsis roots"

We expect to receive your revised manuscript within two weeks. 

*Published Peer Review History*

*Press*

Sincerely,

Ines

--

Ines Alvarez-Garcia

Senior Editor

PLOS Biology

on behalf of

Melissa Vazquez Hernandez, Ph.D.

Associate Editor

PLOS Biology

DATA POLICY:

Many thanks for submitting the data underlying the graphs shown in the figures. We are still missing the data found in the following figures - please add these to the data file:

Fig. 4; Fig. 6A, B; Fig. S9 and Fig. S10

***Please also make publicly available the data you have deposted in NCBI with BioProject number is PRJNA1164981. In additoin, please obtain a doi in Zenodo for the data deposited in Github by following this link:

https://cassgvp.github.io/github-for-collaborative-documentation/docs/tut/6-Zenodo-integration.html

CODE POLICY

---

## [Editor Report · Decision Letter 3]

31 Oct 2024

Dear Dr Dai,

Thank you for the submission of your revised Research Article "Systematic characterization of plant-associated bacteria that can degrade indole-3-acetic acid" for publication in PLOS Biology. On behalf of my colleagues and the Academic Editor, Cara Helene Haney, I am pleased to say that we can in principle accept your manuscript for publication, provided you address any remaining formatting and reporting issues. These will be detailed in an email you should receive within 2-3 business days from our colleagues in the journal operations team; no action is required from you until then. Please note that we will not be able to formally accept your manuscript and schedule it for publication until you have completed any requested changes.

PRESS

Sincerely, 

Melissa

Melissa Vazquez Hernandez, Ph.D., Ph.D.

Associate Editor

PLOS Biology
